# Deep Learning for Daily Monitoring of Parkinson’s Disease Outside the Clinic Using Wearable Sensors

**DOI:** 10.3390/s22186831

**Published:** 2022-09-09

**Authors:** Roozbeh Atri, Kevin Urban, Barbara Marebwa, Tanya Simuni, Caroline Tanner, Andrew Siderowf, Mark Frasier, Magali Haas, Lee Lancashire

**Affiliations:** 1Cohen Veterans Bioscience, New York, NY 10018, USA; 2The Michael J Fox Foundation for Parkinson’s Research, New York, NY 10163, USA; 3Feinberg School of Medicine, Northwestern University, Chicago, IL 60611, USA; 4Department of Neurology, Weill Institute for Neurosciences University of California, San Francisco, CA 94143, USA; 5Parkinson’s Disease Research Education and Clinical Center, San Francisco Veteran’s Affairs Medical Center, San Francisco, CA 94121, USA; 6Department of Neurology, University of Pennsylvania Perelman School of Medicine, Philadelphia, PA 19104, USA

**Keywords:** Parkinson’s disease, wearable sensors, deep learning, human activity recognition

## Abstract

Now that wearable sensors have become more commonplace, it is possible to monitor individual healthcare-related activity outside the clinic, unleashing potential for early detection of events in diseases such as Parkinson’s disease (PD). However, the unsupervised and “open world” nature of this type of data collection make such applications difficult to develop. In this proof-of-concept study, we used inertial sensor data from Verily Study Watches worn by individuals for up to 23 h per day over several months to distinguish between seven subjects with PD and four without. Since motor-related PD symptoms such as bradykinesia and gait abnormalities typically present when a PD subject is walking, we initially used human activity recognition (HAR) techniques to identify walk-like activity in the unconstrained, unlabeled data. We then used these “walk-like” events to train one-dimensional convolutional neural networks (1D-CNNs) to determine the presence of PD. We report classification accuracies near 90% on single 5-s walk-like events and 100% accuracy when taking the majority vote over single-event classifications that span a duration of one day. Though based on a small cohort, this study shows the feasibility of leveraging unconstrained wearable sensor data to accurately detect the presence or absence of PD.

## 1. Introduction

Parkinson’s disease (PD) is one of the most common and fastest growing neurological disorders [1] that results in a progressive decline in both motor (e.g., bradykinesia) and non-motor (e.g., cognition and mood) symptoms. Since there are currently no objective biomarkers in PD [2], diagnosis is complicated and typically involves clinically administered questionnaires such as the Hoehn and Yahr rating system [3] and Unified Parkinson’s Disease Rating Scale (UPDRS) [4] to assess the severity of a range of symptoms across multiple categories. Additionally, the absence of specific and objective metrics of disease progression is a major obstacle to both early diagnosis and monitoring of disease course, hindering our ability to guide treatment in a subject-specific manner [2].

Although the existing approaches to disease diagnosis have shown moderate-to-excellent inter-rater agreement and test–retest reliability in many analyses [5,6,7,8], this is not always the case, and inter-rater variabilities associated with the UPDRS, MDS-UPDRS, and Hoehn and Yahr scales have been shown to result in up to 40% diagnosis error [9]. These more subjective measures can be highly variable as a result of fluctuations arising from the specific cross-sectional snapshot at the time they are measured [10], potentially leading to symptoms being undetected or mischaracterized, which in turn may delay diagnosis or result in a misdiagnosis with a condition similar to PD, such as multiple system atrophy, progressive supranuclear palsy, corticobasal degeneration, and vascular parkinsonism [11,12]. Therefore, given the multifaceted nature, heterogeneity, and fluctuations of this disease both in its early stages and throughout its progression, objective longitudinal measurements of an individual’s motor state and symptoms are required to better understand symptom variability and the trajectory of disease progression.

Sensor technology is developing rapidly, and these devices are becoming an increasingly practical approach to objectively detecting and diagnosing disease. To date, studies have mostly focused on the collection of “lab-recorded” sensor data to evaluate whether there are data signatures (digital biomarkers) associated with the characteristic features of PD (e.g., freezing of gait [13,14,15], tremors [16,17], and changes in gait pattern [18,19]) that are able to objectively and consistently discriminate between a PD population versus healthy controls (HCs).

Previous studies have investigated how similar tools can be used to aid in PD detection and diagnosis. For example, using data collected from activities such as walking and finger tapping, Prince et al. [20] used a sensor-based technique to detect PD with 62% accuracy. In a review, Rovini et al. [21] identified five studies that analyzed sensor-driven signatures associated with gait and postural sway that focused on early detection of PD. Another study showed how placing an accelerometer at the center of mass resulted in a statistically significant difference in postural sway between subjects with idiopathic rapid eye movement (REM) sleep behavior disorder (RBD) and HC [22], suggesting its use as an indicator of PD, given literature estimates that approximately half of those with RBD will go on to develop PD [23,24,25]. Separately, using hip- and head-based accelerometer recordings of gait, Brodie et al. [26] showed a significant difference between HC and PD in gait parameters such as uncontrolled anteroposterior oscillations, lateral head jerk, and gait stability. Using a gait symmetry measure derived from four limb-distributed inertial sensors, several studies have been able to classify PD and HC subjects with high sensitivity and specificity [27,28].

The above studies are important in that they demonstrate the feasibility and efficacy of sensor-based approaches in the quantification of PD for diagnostic and detection purposes. However, while they provide some validation and motivation for using sensor data to assist in automating and objectifying PD diagnosis, their scope has been limited. For example, the sensors used in such studies are often custom made for the study and, in some cases, deployed in multiple body locations. Although these common practices help classify human activity [29], they also add complexity to operationalizing the research for clinical practice compared with the use of more commercialized devices (e.g., Fitbit, Apple Watch, or the Verily Study Watch).

Another major limitation of these studies is that they focused on labelled, lab-recorded data and therefore did not address challenges regarding daily or weekly fluctuations in symptom severity. Because of the complexities of human motion outside the lab, sensor data that has been collected in supervised settings (e.g., lab sessions with expert oversight or the use of an interactive app in a controlled manner) generally cannot be used to train a model that will be deployed on unconstrained, real-world sensor data. In these cases, to maintain model performance during deployment there is an implicit assumption that the model will be administered in settings and conditions that closely resemble the controlled way the training data was collected. Therefore, such lab-based models do not necessarily extend or generalize to a subject’s natural environment and are typically limited to infrequent and irregular intervals (e.g., only during in-clinic visits), leaving significant gaps in the data collection and disease monitoring of PD.

To overcome some of these challenges and to provide personalized objective measures for PD diagnosis and monitoring, analytical and predictive models trained on real-life sensor data are needed. One solution to resolve this limitation and better understand fluctuations in disease progression is to directly measure human activity data passively and continuously via wearable sensors in a subject’s natural environment. This would also enable both physicians and researchers to monitor health-related metrics more efficiently and frequently. Event detection and activity recognition algorithms can be used to semantically enrich and annotate such sensor time series and aid in the extraction of “digital biomarkers” that can serve as proxies for phenomena such as symptom variability and disease progression.

In this study, we explored the potential for predicting and tracking the presence of PD in real-life, wrist-oriented inertial sensor data recorded by the Verily Study Watch. Accelerometer and gyroscope data were collected daily over several weeks at high resolution (100 Hz) in a subject’s natural environment. PD is primarily characterized by alterations in motor-related features, such as bradykinesia (slowness of motion), gait abnormalities, and tremor, all of which can occur during intervals of walking. We have developed a HAR module that identifies intervals where subjects appear to be walking, which we refer to as “walk-like events” (i.e., events during which the frequency component resembles that of known walking intervals). Using these walk-like events, we then extracted discriminatory features from the unlabeled, passively collected inertial time series data. Since motor-related symptoms of PD can manifest in complex ways, their detection can be challenging using rules-based signal-processing techniques like those in the HAR module used above to identify the walk-like events. This led to the use of convolutional neural networks (CNNs) to perform this subsequent step in our learning framework. CNNs learn to extract increasingly complex compositional features over many sequentially organized convolutional layers and have shown excellent utility in tasks such as image processing, object detection, and classification, without the need for a separate feature engineering step [30]. Such utility can be extended to sequence and time series data with little-to-no modification by using 1-dimensional convolutions [31]. Thus, our driving hypothesis was that a CNN should be capable of learning the features present during walk-like events that could be used to discriminate between PD and HC subjects during arbitrary, relatively short walk-like intervals using a subject’s real-life wearable sensor data.

## 2. Data Collection

Data used in the preparation of this article were obtained from the Parkinson’s Progression Markers Initiative (PPMI) database on 18 June 2021 (www.ppmi-info.org/access-dataspecimens/download-data). For up-to-date information on the study, visit ppmi-info.org. PPMI is a longitudinal, observational clinical study launched in 2010 to verify progression markers in Parkinson’s disease and establish a comprehensive set of clinical, imaging, and bio sample data that can be used to discover and define biomarkers of PD progression. The full PPMI dataset contains longitudinal measurements for more than 1700 individuals (as of June 2022) from multiple cohorts of interest (e.g., clinically diagnosed PD, non-manifest gene carriers, healthy controls, and prodromal) and spans more than 30 clinical sites distributed over 11 countries.

In the primary arm of the PPMI study, the recruited PD subjects are those with early (diagnosed within the past two years), untreated PD with (i) asymmetric resting tremor, (ii) asymmetric bradykinesia, or (iii) two of bradykinesia, resting tremor, and rigidity. HC participants must not show any significant neurologic dysfunction, have no first-degree family members diagnosed with PD, and perform above threshold on a cognitive test. Evaluations are performed at baseline and at 3-month intervals over the first year, followed by evaluations every 6 months in subsequent years (see the PPMI website for more detailed information on the inclusion/exclusion criteria, cohorts, timeline, etc. [32]).

More recently (2018), PPMI launched a substudy using the Verily Study Watch at sites within the United States. In this substudy within PPMI, all US-based subjects enrolled in PPMI were invited to participate. Thus, compared to other data types associated with the full PPMI dataset, the wearable data are not restricted to following the progression of PD starting at an early, untreated stage but may begin at any point along the trajectory. For example, several of the PD PPMI-Verily subjects had enrolled in PPMI many years prior and had been on PD medications continuously shortly thereafter. Other PD subjects in the dataset were newly recruited early-PD subjects who remained untreated well into the sensor data collection.

During the PPMI-Verily study, subjects were asked to wear the Verily Watch for up to 23 h per day from several months to two years as they engaged in their daily activities. The average and median daily wear times across subjects were approximately 18 and 21 h per day, respectively; the interquartile range of recording days spanned 254 to 560 days. For at least one hour per day, subjects were asked to charge the device on the Verily StudyHub and to transfer the data to the cloud [33]. The generated Verily Watch data included triaxial inertial sensors (accelerometer and gyroscope) sampled at 100 Hz, as well as other sensors such as a three-channel photoplethysmogram (PPG), electrodermal activity (EDA), and environmental sensors for pressure, temperature, and humidity. The sensor data were collected without supervision or contextual annotations (e.g., sleeping, walking, sitting).

For this proof-of-concept study, we used a small subset of the full PPMI-Verily wearable dataset. This comprised several consecutive months of data from 11 subjects (four of whom were HCs and seven of whom were clinically diagnosed PD subjects—five from the genetic PD cohort and two from the de novo cohort [34]). The genetic PD cohort were people with PD and pathogenic genetic variant(s) in LRKK2, GBA, and SNCA genes. We further restricted the data to those from the accelerometer and gyroscope sensors over a subset of days within the first three months of data collection. Subject characteristics and demographics were mapped to these data using other data types readily accessible via the PPMI website [32]; see Table 1.

## 3. Methods

Figure 1 shows the high-level details for the proposed pipeline. The pipeline consists of four major modules: preprocessing, dynamic activity recognition, walk-like activity detection, and PD detection. To summarize: triaxial accelerometer and gyroscope sensor data collected from free-living subjects in their daily environment is first passed to the preprocessing module where it is down-sampled and windowed. The windows are then piped into the HAR detection module, which consists of several rules-based signal processing techniques designed to identify dynamic activity from the accelerometer data and use dynamic events to characterize how walking events should appear in the gyroscope component of the inertial data. This is a pseudo-labeling procedure used to annotate and contextualize the data; it serves as a pass/reject gate that forks the windows into positive and negative subsets. The positive events are those that resemble walking; they are intentionally referred to as “walk-like” events to denote the inherent uncertainties in classifying unconstrained, open-world data. Only walk-like events proceed to the next pipeline module. At training time, the walk-like events are split into training and validation sets at the subject level using a leave-one-group-out (LOGO) cross-validation (CV) procedure. For each fold, training data is fed into a CNN to learn an end-to-end feature extraction and PD detection module. The holdout fold is used for validation during training. Test data for each subject are held out of the LOGO-CV procedure altogether. The final diagnostic classification at test time is the result of a majority vote taken over multiple single-event classifications that span a duration of one day.

The following subsections present the details of the preprocessing, dynamic activity detection, walk-like event recognition, and PD prediction modules in greater detail.

### 3.1. The Preprocessing Module

The 6-channel inertial time series data from wrist-oriented triaxial accelerometer and gyroscope sensors worn by both PD and HC subjects were down-sampled from 100 Hz to 20 Hz. The Shannon–Nyquist sampling theorem states that a band-limited signal can be perfectly reconstructed from a discrete set of measurements given a sampling rate twice its highest frequency [35]. Therefore, at 20 Hz resolution, spectral signatures in 2–3 Hz should distinguish rest states from motion and determine walking status [36]; Similar optimal and adequate sampling rates for HAR have been investigated and verified [37,38,39,40]. The time series were then segmented into small, non-overlapping windows. The window length of 5 s was chosen empirically.

### 3.2. Dynamic Activity Detection

The second module in the pipeline was used to parse the data based on activity recognition algorithms and to annotate the data with activity pseudo-labels.

An adaptive threshold was calculated for each subject to include the subject variability for the sensor data processing. This was achieved using a sliding, non-overlapping 5-s window, where the mean absolute value for the accelerometer and gyroscope was calculated and half of the maximum value was stored as the threshold. The definition for the threshold was chosen empirically and calculated on a subject-by-subject basis using the data on the first day only for each subject.

The first class of activity determined whether the subject was in motion or at rest (e.g., “activity” vs. “rest” [41] or “dynamic” vs. “static” [42,43]). Operationally, this dynamic activity detection step serves as a pass/reject gate to fork the windows into two subsets, where “dynamic” events pass and “static” events are rejected. The dynamic activity detection algorithm first selects the accelerometer axis with the highest mean absolute value as the dominant axis [44], ensuring that the algorithm is able to consistently detect dynamic activity regardless of sensor placement and orientation. A dynamic event is defined as any 5-s window in the dominant-axis accelerometer data that has a mean absolute value greater than its associated threshold (defined in Step 1 above). Windows failing to meet a threshold are identified as static and rejected, and they do not move on to the next step of the HAR module.

### 3.3. Walk-like Detection

In the next step of the HAR module, the level of semantic resolution was increased by using a frequency-based walk-detection algorithm on the gyroscope data to further classify whether the dynamic candidates are walk-like events or not. First, the dominant axis for the gyroscope is extracted by calculating the mean absolute value of the rotation in each axis and identifying the axis with the highest values. This procedure helps ensure that the walk-like detection algorithm works on inertial data regardless of placement and orientation. Power spectral density (PSD) was estimated using Welch’s method [45]. The PSD is used to calculate the average power in the frequency range of walking, which is from 0.6 to 2 Hz [35]. The power in this range is compared with power in the rest of the PSD, and the event is labeled walk-like if the power in the walking range is higher than the power in the rest of the power spectrum. To ensure that the signal contains enough energy and to add more regularization to the walk-like detection pipeline, a threshold was applied to the average power for walk-like events. The threshold of 100 was chosen based on an exhaustive search carried out on the USC-SIPI Human Activity Dataset (USC-HAD) [46], where various thresholds between 0 and 500 with a step length of 10 were tested.

When a walk-like event is detected in the gyroscope data, the associated 6-channel window (including all three axes from both the accelerometer and gyroscope) is annotated as such; only the walk-like windows are used in the subsequent PD classification module.

### 3.4. Convolutional Neural Network

Walk-like event windows were then used to train a CNN for PD discrimination. Hyperparameter optimization was carried out to determine a satisfactory architecture for PD detection. Figure 2 illustrates the architecture of the proposed CNN for PD diagnosis using walk-like events. The CNN architecture chosen consisted of four convolutional layers, each with a 5-point kernel applied (stride of 2 and no pooling). The 2-point stride caused the temporal resolution of the channels to decrease by half at each layer. However, the “receptive field” (RF) captured at each time step multiplicatively increases as a function of stride and kernel size [47], and (by design) the number of channels increases as well (8, 16, 32, 64).

Several regularization techniques were used during training to avoid overfitting and promote generalizability [30]. L2 regularization was used at each convolutional layer, followed by batch normalization [32,33] and ReLU activation. During training, a dropout layer with a 50% dropout rate was also included after the final convolutional layer (dropout can slow down training but pushes the network to robustly learn generalizable features in a distributed, “multi-path” fashion [31]).

The output of the last convolutional layer was flattened and SoftMax activation was used for binary classification. Gradient descent using the Adam optimization algorithm [48] was used with a binary cross-entropy loss function. The network was implemented using the Keras API from the TensorFlow-GPU package (version 2.1) in Python; other tuning parameters not explicitly mentioned were set as default.

### 3.5. Comparison with Other Machine Learning Approaches

We selected a CNN as the basis of our PD detection classifier with a hypothesis that a CNN can automatically learn which features within the 5-s walk-like event time series are most optimal in discriminating between HC and PD subjects. To assess and contextualize this hypothesis, we also removed the CNN module from the pipeline and replaced it with a more traditional machine learning module, including separate feature extraction and classification submodules.

In the modified pipeline, the feature extraction submodule is no longer learned at training time as it is with the CNN in the original pipeline. Instead, the extraction of features must be prespecified in a rules-based fashion, a significant departure from the hypothesis underlying the CNN module. With a CNN, the number of features is determined by the width of the dense layer preceding the output layer. For any given number of features, N, the CNN will learn a non-unique but optimized set of N features. Here, N itself can also be optimized during a hyperoptimization procedure without any further specifications or human interjection. These properties are lost in the modified pipeline where features are predefined prior to training without any promise that they include all the relevant information contained within the time series.

In order to compare our CNN module, for both the accelerometer and gyroscope we defined the following features across the segmented non-overlapping 5 s windows:-Mean for each x, y, z axis-Variance for each x, y, z axis-Skew for each x, y, z axis-Kurtosis for each x, y, z axis-Maximum across axes-Total low-frequency band power from acc (0–4 Hz; power for each x, y, z axis)-Total mid-frequency (tremor range) band power from acc (4–9 Hz; power for each x, y, z axis)-Total high-frequency band power from acc (9+ Hz; power summed for each x, y, z axis)

To compute band power quantities for a 5-s walk-like event, we applied a Hanning taper to the event window prior to FFT, then computed the square modulus at each frequency; a given band power quantity was then defined as the sum of power over the given frequency range. These specific ranges were selected to capture Parkinsonian tremor which is defined as a rest tremor or rest and postural tremor [49]. Here, rest tremor is above 4 Hz, while isolated postural and kinetic tremors may vary between 4 and 9 Hz. High-frequency tremors between 8 and 12 Hz may also present in PD.

We then used the above features to train a range of machine learning classifiers. Here, we selected logistic regression, random forest, gradient-boosted trees, and elastic net. All models were trained using sci-kit learn in Python, setting parameters as default as with the CNN model.

### 3.6. Model Performance

Given limited subjects in the current study, we chose to use leave-one-group-out (LOGO) cross-validation (CV) to measure model performance, where each subject is treated as a group and excluded completely in one of the training folds and then used to assess model performance in the validation fold. Although computationally expensive, this has been shown to be a robust approach in settings with a small sample size [50].

In addition to the LOGO strategy and to further test the derived models, the full training/validation dataset was limited to the first 10 days of sensor data collection for each subject. Held-out sensor data that was collected in the future of this training/validation dataset was then used to mimic model deployment and validate the out-of-sample temporal stability as well. Thus, for each subject-level fold, a model was trained using only the data from the first 10 days of the fold’s training subjects, validated on the first 10 days of the fold’s validation subject, and further tested on the validation subject’s holdout data recorded up to three months after their validation data.

## 4. Results

Table 1 shows general demographic information, as well as the gait score from the UPDRS examination closest to the sensor data collection. During UPDRS examination, gait and other motor symptoms are observed and rated by an expert. Specifically, for gait, subjects walk toward the examiner who observes the gait on both sides of the subject. Various gait metrics are observed, such as stride amplitude, stride speed, height of foot lift, heel strike during walking, turning, and arm swing [51]. The score ranges from 0 to 4 with integer steps, which respectively indicate normal, slight, mild, moderate, and severe. As expected, the HC all scored zero, which means there are no problems in their gait pattern. From the PD cohorts, three subjects scored zero (normal) and three subjects scored one (slight), which is described as “independent walking with minor gait impairment”. Only one subject from the de novo PD cohort scored two, which indicates mild gait issues.

Ten consecutive days of raw sensor data from each subject were processed using the preprocessing and HAR pipeline modules (described above) to extract walk-like events. Figure 3 visually contrasts the temporal characteristics (in normalized units) of a single 5-s event as measured by the triaxial accelerometer and triaxial gyroscope, in addition to illustrating a sample of a detected walk-like event (highlighted) juxtaposed between non-walking events. The walk-like event-detection pipeline was validated using USC-HAD, the results of which showed that the proposed walk-like detection algorithm precisely detected walking events with over 94% accuracy on these in-lab, closed-world datasets.

Figure 4 shows the distribution of daily-integrated walk-like event durations grouped by diagnosis and for each subject individually. Figure 4a highlights the mean daily walking duration for HC being significantly higher compared with PD subjects (5.65 h and 4.3 h, respectively; *t*-test *p* < 0.001), while more variance was observed in the PD subjects with some PD subjects showing very limited movement on average (Figure 4b).

The down-sampled inertial data windows annotated as walk-like events were used to train a separate binary classifier (PD vs. HC) for each subject by the majority vote from each walk-like event aggregated over each day. Figure 5a shows the LOGO-CV results for the first 10 days using both accelerometer and gyroscope data (Acc + Gyro), accelerometer only, and gyroscope only, where the average classification accuracy for single walk-like events at the group level was 98.5% (±3.6%) and 88.7% (±17.2%) for HC and PD subjects, respectively. In most cases, using both the accelerometer and gyroscope led to improved classification accuracies, although these were typically overlapping in confidence intervals. The final aggregated model, defined by taking a majority vote over multiple walk-like events within a single day, achieved 100% accuracy.

The results of the classification for the subjects after one, two, and three months in the future were similar in that the average single-event accuracy was 99.2% and 90.2% for HC and PD subjects, respectively (Figure 5b). The average single-event accuracy per day per subject was above 50% for all subject-days. Thus, by taking a daily majority vote, all subjects were correctly classified on every day tested (100% classification accuracy). 

Table 2 shows the LOGO-CV results for the first 10 days using the combined accelerometer and gyroscope data across the other machine learning classifiers using the extracted features described above. Although average classification accuracy was similar across the methods, the CNN outperformed all methods when predicting PD subjects, indicating improved sensitivity when using the CNN approach. The closest performance was seen using Elastic Net, which was approximately 3% less accurate here.

## 5. Discussion

The lack of labelled wearable sensor data in unconstrained, free-living contexts is a major obstacle in developing reliable wearable-based subject-monitoring systems. The diversity and uncertainty of human activity throughout the day introduces challenges with respect to sensor integration and the utility of these devices in health monitoring. Hence, it is of high importance to consider such uncertainties when developing algorithms for example applications such as monitoring human activity or detecting presence or absence of diseases such as PD. In this study, we introduced a pipeline that applies techniques from signal processing and deep learning to analyze, segment, and classify unconstrained inertial sensor data from the Verily Watch. Here, data were continuously recorded in a passive manner outside of the clinic (natural settings) for up to 23 h per day and did not include contextual annotations (e.g., activity labels). The importance and major contributions of this work arise from the practical value of such uncontrolled, real-world wearable data, where there is no limitation to the number, variation, or complexity of activities a subject may engage in throughout their daily life.

Rules-based signal-processing techniques were used to distill “walk-like” events from the data. First, dynamic activity detection was performed by thresholding the accelerometer data in a manner that was adaptive to different sensor placements and orientations. The gyroscope sensor was used for this purpose because of its sensitive nature and cyclic behavior during walking [44]. To overcome challenges with sensor placement location and orientation, a dominant-axis identification pipeline was used to detect the dominant axis, hence reducing the dependency on standardized sensor placement. The PSD for the dominant axis of the gyroscope was estimated and the average power over the walking frequency range (0.6–2 Hz) calculated. Walk-like events were detected if the walking frequency range had a higher-than-average power. The term “walk-like” was used for such events to emphasize the uncertainty of the approach, which will detect other activities in the same frequency range. However, our hypothesis was that this algorithm uses rules that are capable of selecting out events in a repeatable, deterministic manner that are likely to be walking events but, more importantly, are shown to carry discriminatory signals that enable PD classification. The walk-like detection algorithm was validated on publicly available, labelled datasets and the results confirmed the hypothesis with high accuracy (94%).

Once the walk-like events were identified, they were used to train a CNN for PD classification. Unlike classical classification approaches that require subjective feature engineering, CNNs are capable of objectively learning relevant features for classification with little or no feature engineering. Though originally used for classification tasks on images, which consist of two-dimensional data channels, CNNs are also useful for time series and sequence data types, which are composed of one-dimensional channels. CNN learns the internal representations of the data without relying on expert domain knowledge and can achieve comparable or better results.

In addition, restricting the CNN inputs to walk-like events reduces the complexity of what the CNN must learn during training, making it easier for the deep learning algorithm to learn and extract features that discriminate between PD and HC subjects. Because of the limited number of subjects in this study, a subject-level LOGO-CV scheme was applied to make most effective use of the data, as well as to prevent data leakage and subject bias issues (subject-level training folds ensure that CNN is not exposed to behaviors or patterns specific to the test subject). This cross-validation method generates subject-level folds such that each subject is treated as the lone test subject in one of the folds (specifically, in the fold that excludes all data from that subject in the training subset). Hence, this method makes efficient use of the subject data and helps provide validation for the data processing and modeling approach in general by providing validation estimates on trained instances for each subject-level fold.

The daily single-event accuracy averaged at approximately 99% for HCs and 89% for PD subjects, and this performance was maintained when predicting using unseen data up to 3 months out. A higher-confidence diagnosis was reported daily for each subject by computing the majority vote over each day’s walk-like events. Because of the already high single-event accuracy, this scheme resulted in perfect diagnostic classification accuracy on all subject-days tested. The high classification accuracy suggests that the deep learning method applied here learns to extract general differences in motor-related features between HC and PD populations.

Given that subject-specific walk-like behaviors might change with time and disease progression, data collected one, two, and three months after the training/validation interval for each subject was also used to test the model’s prediction accuracy. Single-event classification accuracy remained high; importantly, 100% correct classification was observed daily for all post-training subject-days tested when using the majority vote. This finding indicates that the CNN was able to identify stable discriminatory features from the walk-like events that remain useful for at least several months without any negative deviations or drift in model quality metrics. Importantly, the method outperformed other popular machine learning based classifiers while offering the additional advantage of not requiring any specific domain expertise when selecting and engineering features from the sensor data, thereby learning the optimal combination of features in an automated and unbiased fashion.

The proposed algorithm showed an accuracy of 100% for daily PD diagnosis based solely on walk-like events, which can be interpreted as the ability to identify subtle gait changes related to PD that are unaccounted for in the UPDRS scores. This can be clearly seen from genetic PD subjects that have normal gait in their UPDRS exam. In practice, this might be a result of limited observation or shortcomings of self-reporting. PD symptoms tend to change on different timescales—including hourly and daily variations—and are driven by variables related to sleep hygiene, food intake, medication, etc., which may limit objectivity of the clinical analysis [52,53,54].

This proof-of-concept study using sensor data from daily activity shows the potential for using free-living, wearable sensor data that may improve monitoring in diseases such as PD by capturing measurements more frequently. Future directions will focus on identifying and quantifying specific PD symptoms in larger samples and introducing daily, weekly, and monthly summary metrics. This may also provide a better understanding of the disease (e.g., high-resolution medication effects on symptom severity) and subject behaviors (e.g., treatment plan compliance). These techniques may enable us to complement observation-based instruments, such as the UPDRS, that are captured less frequently, by augmenting highly specialized domain expertise and assisting non-specialist or relatively inexperienced clinical personnel in diagnostic decision making. It may also help reduce in-clinic expenses and mitigate various scheduling and time commitment burdens on subjects, caregivers, and clinicians [8,55,56].

## 6. Conclusions

The current study investigated the possibility of using a commercial smart watch to passively and continuously collect activity data from subjects in their natural environment on a daily basis without constraints and using the data to detect the presence or absence of PD across a range of heterogenous mobility and disease severity scores. Promising results are presented, which can be used in objective PD diagnosis and may pave the way in determining more specific and objective metrics for monitoring trajectories of disease progression as well as improving clinical decision making and guiding therapeutic interventions. Future work will explore the utility of this approach using the full PPMI-Verily dataset where we will test against more distant time points post-training (e.g., at 6, 9, and 12 months).

## Figures and Tables

**Figure 1 sensors-22-06831-f001:**
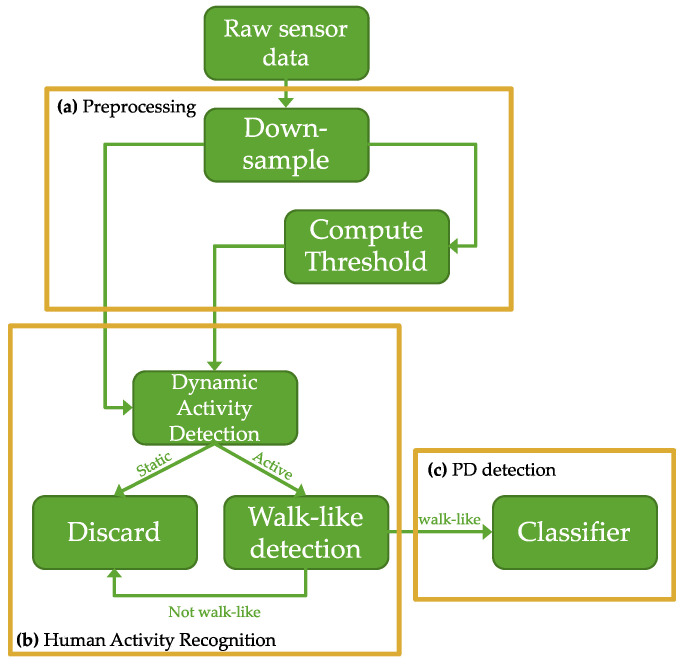
Block diagram of sensor-based PD detection pipeline: (**a**) preprocessing pipeline to down-sample the inertial data and calculate an adaptive threshold; (**b**) human activity recognition pipeline to parse the inertial data to dynamic activity and identify walk-like events from them; (**c**) deep convolutional neural PD classifier.

**Figure 2 sensors-22-06831-f002:**
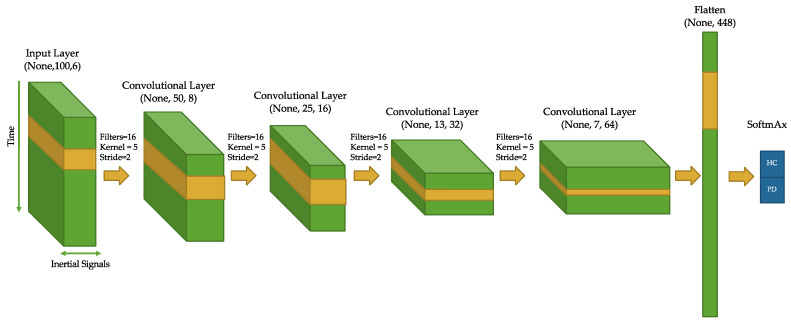
Architecture of the one-dimensional convolutional neural networks classifier for PD detection using 6-channel inertial data from walk-like events.

**Figure 3 sensors-22-06831-f003:**
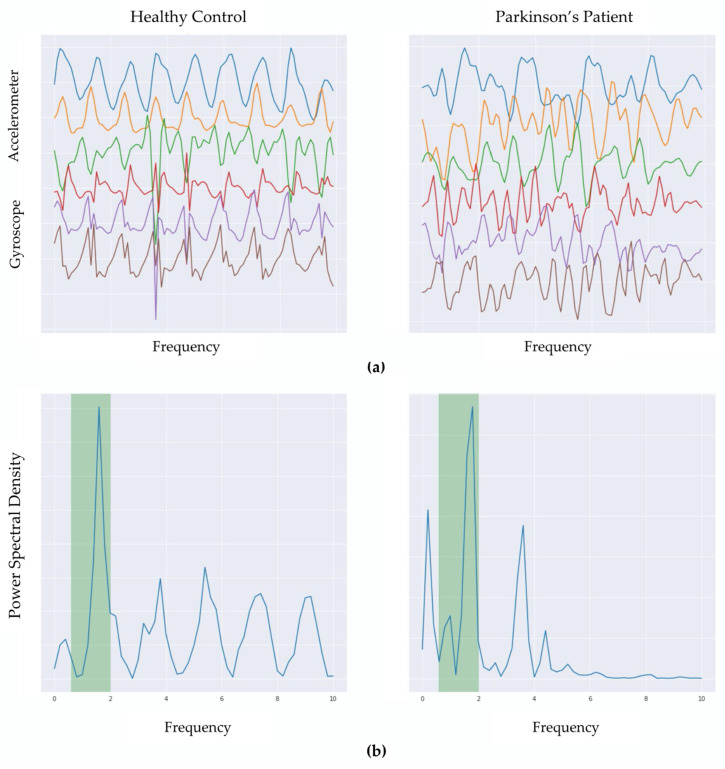
(**a**) Temporal characteristics (in normalized units) of a single 5-s event as measured by the triaxial (x, y, z axis) accelerometer (top 3 traces) and triaxial gyroscope (bottom 3 traces). These examples illustrate an identified walk-like event for HC (left) and PD (right). (**b**) Frequency spectrum power (power spectral density) of these 5-s windows, with the highlighted column illustrating that a sinusoidal frequency within the walking range (0.6–2 Hz) dominates the frequency power.

**Figure 4 sensors-22-06831-f004:**
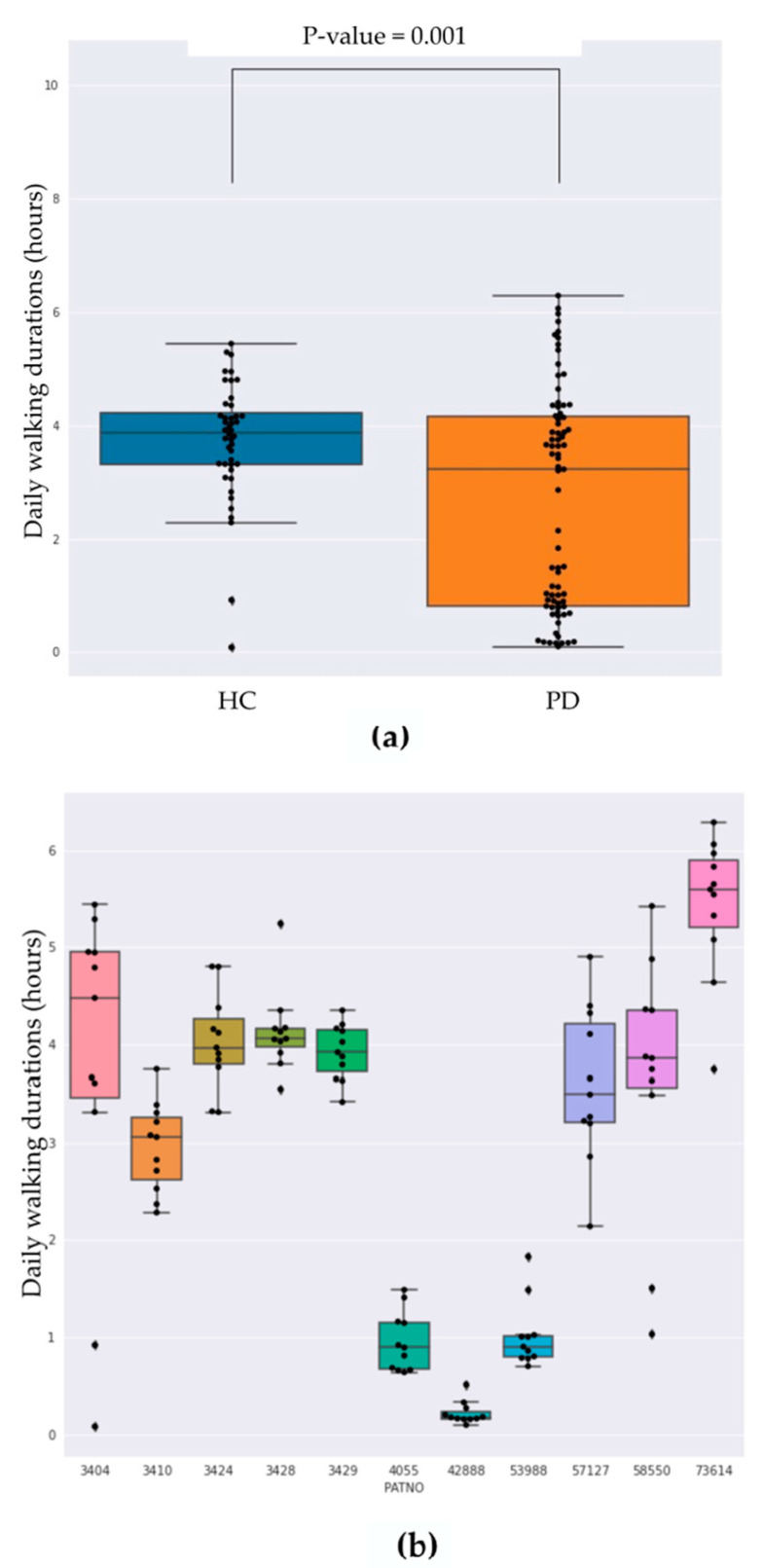
(**a**) Box plots showing the distributions of daily walk-like durations over the first 10 days of data collection summarized at the cohort level. (**b**) Box plots for each individual subject where the first five subjects are defined as healthy controls and the remaining are subjects with Parkinson’s disease. For both plots, whiskers indicate minimum and maximum values, excluding outliers.

**Figure 5 sensors-22-06831-f005:**
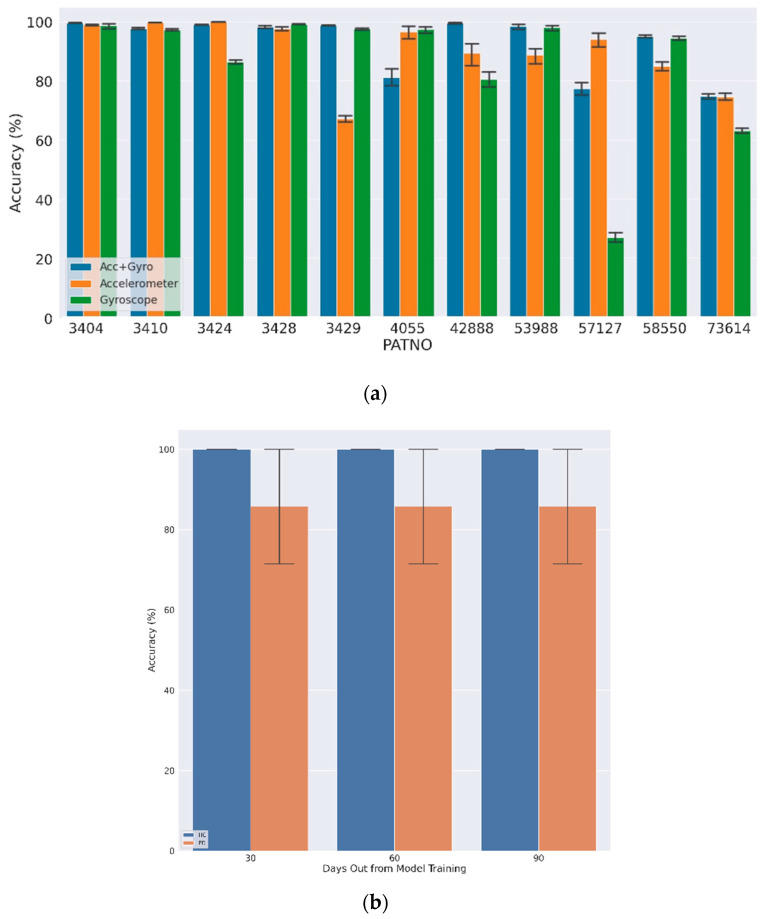
Daily average PD classification results: (**a**) average daily LOGO CV accuracy for first 10 days of sensor data; (**b**) average 5-day accuracy for data after 1, 2, and 3 months.

**Table 1 sensors-22-06831-t001:** PPMI participant characteristics.

Subject	Sex	Age	Diagnosis	Years Since PD Diagnosis	Years on PD Medication	Dominant PD Side *	Wrist (Sensor)	Dominant Hand	MDS-UPDRS NP3GAIT Score	MDS-UPDRS3 TOTAL Score **
3404	F	64.0	HC	-	-	-	R	L	0	0
3410	M	82.0	HC	-	-	-	L	R	0	0
3424	F	71.2	HC	-	-	-	L	R	0	3
3428	F	65.4	HC	-	-	-	L	R	0	1
3429	M	72.4	De Novo PD	7	3.9	R	L	R	0	23
4055	M	75.9	De Novo PD	6	4.5	L	L	L	1	33
42888	M	69.2	Genetic PD	1	1.6	S	L	R	0	33
53988	M	58.1	Genetic PD	8	2.5	R	R	R	1	13
57127	M	77.6	Genetic PD	4	2.7	L	L	R	0	22
58550	M	73.9	Genetic PD	8	1.0	R	R	R	1	25
73614	F	76.5	Genetic PD	3	N/A	R	R	R	2	52

* S: symmetric, L: left, R: right, ** data as of final visit in database.

**Table 2 sensors-22-06831-t002:** Comparison of model performance with other machine learning classifiers.

Method	Healthy Control Mean Accuracy	Parkinson’s Disease Mean Accuracy
**CNN**	**99%**	**90%**
Logistic Regression	99%	64%
Random Forest	99%	85%
Gradient-Boosted Trees	99%	72%
Elastic Net	100%	86%

## Data Availability

Data and study protocols can be found on the PPMI website and portal (ppmi-info.org). Data used in the preparation of this article were obtained on 18 June 2021.

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
