# Peer review of "Deep Learning for Daily Monitoring of Parkinson’s Disease Outside the Clinic Using Wearable Sensors"

_sensors, 2022, doi:10.3390/s22186831_

Round 1

Reviewer 1 Report

Authors claimed that they introduce Deep Learning for Daily Monitoring of Parkinson's Disease Outside the Clinic using Wearable Sensors. They used inertial sensor data from Verily Study Watches worn by individuals for up 22 to 23 hours per day over several months, to distinguish between subjects with (N=7) and without 23 (N=4) PD. This work reported classification accuracies near 90% on single 5-second walk-like 28 events and 100% accuracy when taking the majority vote over multiple single-event classifications. The topic seems interesting, and author present their findings well. I suggest for some minor correction based on the following points.

1.       In abstract N=7 and N=4 is not clearly understood able. Author should revise this sentence to make is more general to understand the reader who are not very familiar with this topic.

2.       In figure-3, there is no unit of scale in x and y-axis. Moreover, Y axis has no scale. So, it is not clearly understandable.

3.       Authors can present a performance comparison of proposed method with others. 

Author Response

We would like to thank Reviewer 1 for their helpful comments. To address their concerns we have corrected the N’s in the abstract, and provided additional description of the figure in both text and caption to increase clarity. On the point around a comparison of other machine learning approaches, we believe that this is beyond the scope of this manuscript and the significant work required would more likely be suitable as a standalone publication. We would like to emphasize that the purpose of the manuscript was not to compare methods but to show the proof of concept that we can use raw waveforms to classify subjects rather than needing to rely on requiring often difficult to acquire labelled derived data from wearable sensors.

Reviewer 2 Report

In the present article Atri et al. demonstrated the daily monitoring of Parkinson’s disease using the wearable sensor. Following issues needs to be addressed in order to make it published:

1.      Author should provide the comparison table with different method to detect the Parkinson disease.

2.      In figures please indicate that, how many subjects has been considered for test.

3.      In my opinion, author should take other method also in order to prove the reliability of present senor.

4.      If possible, please add data of patients with low and high level of disease in order to prove the efficiency o sensor.   

Author Response

We would like to thank Reviewer 2 for their helpful comments. To address their concerns we have added the following for clarity ‘At training time, the walk-like events are split into training and validation sets at the sub-ject level, using a leave-one-group-out (LOGO) cross-validation (CV) procedure. For each fold, training data is fed into a CNN to learn an end-to-end feature extraction and PD detection module. The holdout fold is used for validation during training. Test data for each subject is held out of the LOGO-CV procedure altogether.’ We have also added an additional column in Table 1 highlighting disease severity, and produced a plot showing individual daily walking duration (Figure 4b).  On the point around a comparison of other machine learning approaches, we believe that this is beyond the scope of this manuscript and the significant work required would more likely be suitable as a standalone publication. We would like to emphasize that the purpose of the manuscript was not to compare methods but to show the proof of concept that we can use raw waveforms to classify subjects rather than needing to rely on requiring often difficult to acquire labeled derived data from wearable sensors.

Reviewer 3 Report

The paper presents an efficient approach to collect data about patients using wearable sensors. These data are then used for machine learning algorithms to predict Parkinson's disease.

The quality of the document is good, only very few mistakes are present, which can be easily corrected.

lines 52-53: Vascular Parinsonism

typo

line 57: While still in their infancy...

This is a quickly developing area with useful good quality sensors. "infancy" is not a proper word here.

line 99: these challenge

this challenge or these challenges

lines 213-214: ... signal can be perfectly reconstructed from a discrete set of measurements given a sampling rate twice its highest frequency ...

very simple statement, not mentioning the conditions

line 224: Dynamic activity decetion

typo

capitals inconsistent with other subsections

fig 5.: very small X axis font size

The presentation of the results is poor.

We need more information about the executions and about the results.

line 29: 100% accuracy when taking the majority vote over multiple single-event classifications

line 201: result of a majority vote taken over multiple single-event classifications

multiple = how many?

I find the conclusions section too short, we need more precise conclusions in this manuscript.

Author Response

We would like to thank Reviewer 3 for their helpful comments. To address their concerns we have corrected the typos and inconsistencies with formatting that were pointed out to us. We have modified the statement around sensors being in their infancy to more accurately reflect the current state of play in this area. We also re-wrote the sentence describing the Shannon-Nyquist sampling theorem to clarify that we meet the condition since we filter out the signal. In order to provide more information around the results, we have added in additional text to the methods, results and conclusions, and more Figures to present these more clearly and comprehensively. We also described the majority vote procedure more clearly to explain how we performed the cross validation procedure to evaluate our model performance.

Round 2

Reviewer 2 Report

In response letter authors have mentioned that "purpose of the manuscript was not to compare methods ". However, comparison here asked to prove that sensor developed by authors exhibits the potential advantage over other already sensors.

So, in my opinion authors should definitely compare their sensor with already existing related sensors.

Author Response

Many thanks for your comments. In recognition of your feedback we are also adding some comparisons with other ML techniques in the latest revision. 

Reviewer 3 Report

The quality of the paper improved a lot.

The authors addressed the issues presented in the review.

Some of the figures have no reason to be in bitmap form in, it could be vector graphic. Vector graphic format has advances like smooth zooming possibility and searchable text.

Some figure axis label font sizes are very small.

Author Response

Many thanks for your comments and we are pleased that you feel the manuscript has been improved. We are also adding in some comparisons with other ML techniques in the latest revision.